# The Scaffold Immunophilin FKBP51 Is a Phosphoprotein That Undergoes Dynamic Mitochondrial-Nuclear Shuttling

**DOI:** 10.3390/cells11233771

**Published:** 2022-11-25

**Authors:** Nadia R. Zgajnar, Cristina Daneri-Becerra, Ana Cauerhff, Mario D. Galigniana

**Affiliations:** 1Instituto de Biología y Medicina Experimental (IBYME)-CONICET, Buenos Aires 1428, Argentina; 2Departamento de Química Biológica, Facultad de Ciencias Exactas y Naturales, Universidad de Buenos Aires, Buenos Aires 1428, Argentina

**Keywords:** immunophilin, FKBP51, phosphoamino acid, nuclear-mitochondrial shuttling

## Abstract

The immunophilin FKBP51 forms heterocomplexes with molecular chaperones, protein-kinases, protein-phosphatases, autophagy-related factors, and transcription factors. Like most scaffold proteins, FKBP51 can use a simple tethering mechanism to favor the efficiency of interactions with partner molecules, but it can also exert more complex allosteric controls over client factors, the immunophilin itself being a putative regulation target. One of the simplest strategies for regulating pathways and subcellular localization of proteins is phosphorylation. In this study, it is shown that scaffold immunophilin FKBP51 is resolved by resolutive electrophoresis in various phosphorylated isoforms. This was evidenced by their reactivity with specific anti-phosphoamino acid antibodies and their fade-out by treatment with alkaline phosphatase. Interestingly, stress situations such as exposure to oxidants or in vivo fasting favors FKBP51 translocation from mitochondria to the nucleus. While fasting involves phosphothreonine residues, oxidative stress involves tyrosine residues. Molecular modeling predicts the existence of potential targets located at the FK1 domain of the immunophilin. Thus, oxidative stress favors FKBP51 dephosphorylation and protein degradation by the proteasome, whereas FK506 binding protects the persistence of the post-translational modification in tyrosine, leading to FKBP51 stability under oxidative conditions. Therefore, FKBP51 is revealed as a phosphoprotein that undergoes differential phosphorylations according to the stimulus.

## 1. Introduction

Immunophilins are proteins belonging to a subfamily of the molecular chaperone family [1,2]. Immunophilins exhibit two distinctive features: (a) they show peptidylprolyl-(cis/trans)-isomerase activity, i.e., the reversible cis/trans interconversion of Xaa-Pro bonds, this being the signature characteristic of the subfamily; (b) they also bind immunosuppressive drugs such as FK506, rapamycin, or cyclosporine A, an interaction that abolishes the enzymatic activity of isomerase. The low molecular weight immunophilins CyPA [3] and FKBP12 [4] are the only responsible agents of the immunosuppressive action of these drugs due to the inhibition of calcineurin, a Ser/Thr-phosphatase belonging to the PP2B type. Larger members of the immunophilin subfamily also show additional domains, among them sequences of 34 amino acids repeated in tandem known as TPR (tetratricopeptide repeats), through which immunophilins interact with Hsp90. Not surprisingly, these Hsp90-binding cochaperones were first discovered associated with Hsp90 client factors such as steroid receptor heterocomplexes—i.e., FKBP51, FKBP52, CyP40, and two immunophilin-like proteins, PP5 and FKBPL [5]. Regarding the scaffolding nature of the Hsp90•immunophilin complex, it should be emphasized that, with the exception of FKBP51 [6], all those immunophilins that have been recovered associated with steroid receptors are capable of binding the dynein/dynactin motor complex via their PPIase domain [7]. As a consequence, all these immunophilins are related to the transport mechanism of the client receptor toward the nucleus. Importantly, it is regarded that under physiologic conditions, there is only one TPR acceptor site functionally available per dimer of Hsp90, a property that shows functional relevance because different TPR proteins compete with one another for the binding to the chaperone in a mutually exclusive manner [8,9,10]. This implies the existence of heterocomplexes of the client factor that show different biological properties according to the nature of the interacting TPR factor. Two of the best-characterized TPR-domain immunophilins are FKBP51 and FKBP52 [11,12], which share ~60% homology and ~75% similarity of sequence and a close-related structural organization [11]. Both immunophilins differ from the biological perspective—while FKBP52 facilitates the active retrotransport of GR on cytoskeletal tracks via dynein. This movement is antagonized by FKBP51 [9], an immunophilin that does not bind the motor protein [6]. Moreover, FKBP52 is required for the transcriptional activity of GR [6], PR [13], or NF-kB [14], whereas FKBP51 frequently shows inhibitory actions. Recently, our laboratory also evidenced that both processes of neurodifferentiation and neuroregeneration are favored by FKBP52 activity and impaired by FKBP51 [15,16], a phenomenon being triggered by the immunophilin ligand FK506.

The recent discovery that a significant pool of FKBP51 is mitochondrial [17,18,19,20,21,22] was quite surprising since this immunophilin does not exhibit a known mitochondrial localization signal. In line with this, it has recently been proposed that the Hsp90/Hsp70 and FKBP51 chaperone complex associated to the conventional mitochondrial translocation machinery are regulators of GR import into mitochondria, thus influencing mitochondrial function and cell survival [23]. An interesting property of FKBP51 is to show antiapoptotic action when it is overexpressed. This agrees with earlier observations that FKBP51 prevents cytochrome c release from mitochondria preventing caspase activation [17,24,25]. Not surprisingly, FKBP51 is overexpressed in several cancer cells [21] and enhances telomerase enzymatic activity when it is recruited to the catalytic subunit of telomerase, hTERT, which is also an Hsp90 client factor [21]. This is an essential contribution of FKBP51 to the rapid clonal expansion of some types of cancer cells. Several pieces of evidence suggest that FKBP51 acquires a pro-oncogenic potential [26,27]; for example, it positively regulates melanoma stemness and metastatic potential [28], it is thought to be a key factor in the progression and chemotherapeutic response of pancreatic adenocarcinoma [29], and it is closely related to acute lymphoblastic leukemia and several variants of breast, ovary, and lung tumor pathologies [27]. FKBP51 is also highly expressed in glioblastomas [30].

The broad spectrum of action of FKBP51 is possible due to the ability of FKBP51 to form several types of heterocomplexes with other chaperones, kinases, phosphatases, autophagy-related factors, transcription factors, receptors, etc. Like most scaffold proteins, FKBP51 is itself a target to be regulated. From this perspective, perhaps one of the simplest evolutionary strategies for regulating pathways and subcellular localization of proteins is phosphorylation. The observed resolution of FKBP51 into several isoforms after a high-resolution electrophoresis and the disappearance of the slower bands (i.e., those with apparent larger molecular weight) after a treatment of cells extracts with alkaline phosphatase led to the thought that FKBP51 shows the ability to be phosphorylated [17]. In this study, we have identified two phosphorylation residues of FKBP51 that are related to biological responses such as cell stress stimulation and cell differentiation, and also to FKBP51 stability.

## 2. Materials and Methods

### 2.1. Reagents

Bovine intestinal alkaline phosphatase (type VII-NT), hydrogen peroxide, MG132, and the mouse monoclonal IgG against the *flag*-peptide (M2 clone) were purchased from Sigma-Aldrich (St. Louis, MO, USA). FK506 was from LC Laboratories (Wobrun, MA, USA). Secondary antibodies labeled with Alexa Fluor Dyes (488, 546, and 647), and MitoTracker-633 dye were purchased from Molecular Probes (Eugene, OR, USA). HRP-conjugated goat anti-rabbit and HRP-conjugated donkey anti-mouse were from Pierce (Waltham, MA, USA), Mouse monoclonal IgGs against Cox-IV, Tom20, and actin were from Abcam (Cambridge, UK). Goat anti-lamin B antibody was from Santa Cruz Biotech (Santa Cruz, CA, USA). Rabbit monoclonal IgG anti-FKBP51 was from Affinity BioReagents (Golden, CO, USA). The MG19 mouse monoclonal IgG anti-FKBP51 and the purified R51 rabbit antiserum were produced in the laboratory [31]. The N27F3-4 anti-72/73-kDa heat-shock protein monoclonal IgG (anti-Hsp70) was from StressGen (Ann Arbor, MI, USA). The antibodies against phospho-amino acids were from Cell Signalling (Danvers, MA, USA). The plasmid encoding *flag*-tagged human FKBP51 (pCIneo-*flag*-hFKBP51) was a generous gift from Dr. Theo Rein (Max Planck Institute of Psychiatry, Munich, Germany). The point mutations of tyrosine-54 and threonine 26 into alanine to generate the Y54A and T26A *flag*-tagged constructs were achieved by Creative Biolabs (Shirley, NY, USA) using the Phusion Site-Directed Mutagenesis Kit from ThermoFisher Scientific- cat#F541 (Waltham, MA, USA) following the manufacturer’s instructions. Restriction enzymes were from Promega (Madison, WI, USA). Oligonucleotides primers were obtained from IDT Technologies (Coralville, IA, USA) and PCR amplification was performed with Pfx50 Taq polymerase (Thermo Fisher Sci, Waltham, MA, USA). Constructs were independently confirmed by nucleotide sequencing (Macrogen, Republic of Korea). The siRNAs for FKBP51 and the scrambled siRNA control were purchased from Thermo Scientific Dharmacon (Lafayette, CO, USA).

### 2.2. Cell Transfection and Cell Lysates

L1-3T3 mouse fibroblasts and N2a mouse neuroblastoma calls were grown in high glucose DMEM (Thermo Fisher Sci, Waltham, MA, USA) supplemented with 10% calf bovine serum (Internegocios, Mercedes, Argentina), 2 mM glutamine, 50 U/mL penicillin, and 50 mg/mL streptomycin (all of them purchased from Sigma). Transfections of pCINeo vectored plasmids encoding for *flag*-tagged human FKBP51 (wild-type or the Y54A or T26A mutants) were performed when cells reached ~40–50% confluence using the TransFast reagent (Promega, Madison, WI, USA). Experiments were performed 35 h post-transfection. Lysates were made by Dounce homogenization in TEGM buffer (TES at pH 7.6, 50 mM NaCl, 4 mM EDTA, 10% (*v*/*v*) glycerol, 20 mM Na_2_O_4_Mo) supplemented with 1 mM NaF, 10 mM Na_2_O_4_V, and one tablet of Complete-Mini protease inhibitor mixture from Roche Diagnostics (Mannheim, Germany) per 2 mL of solution. Immunoprecipitations were achieved by mixing 300 mL whole cell lysates, 2 mL anti-*flag* mouse IgG (clone M2) (or 2 mL of a non-immune IgG), 15 mL protein A-Sepharose (Sigma) followed by rotation for 2.5 h at 4 °C. Pellets were washed three times with 1 mL of TEGM buffer and proteins were resolved in a cold room by 9% PAGE/SDS (75V, ~1 h) for standard runnings, or 15% PAGE/SDS (10 V, overnight) for resolutive runnings. Proteins were transferred to Immobilon-P membranes and Western blotted with the proper primary antibodies followed by HRP-conjugated counter-antibodies. Proteins were visualized by ECL with a BioRad kit (Hercules, CA, USA). Semi-quantifications of the protein bands were performed with the ImageJ program (NIH).

### 2.3. Organotypic Brain Slice Cultures 

Frontal cortex slices of 250 μm from C57BL/6 mice were generated with a Mcllwain tissue chopper. Brain slices were transferred onto plates containing 4% soft agar and culture medium (50% MEM with Earl’s slats and glutamine, 25% Hanks medium, 25% horse serum supplemented with 20 mM Hepes, and 6 mg/mL D-glucose (Gibco-BRL)). The enzymatic activities of CPK (creatine phosphokinase), LDH (lactic dehydrogenase), and GOT (glutamate oxalacetate transaminase) were measured in the culture medium using a commercial kit from Boeringher Ingelheim GmbH Argentina.

### 2.4. Subcellular Fractionation

Untreated cells and cells incubated with 0.25 mM H_2_O_2_ underwent subcellular fractionation into nuclei, cytosol, and mitochondria as it was described in a previous study [21]. The subcellular fractionation from mouse liver was performed according to Fernández–Vizarra [32] using C57BL/6 males (~15–20 g) fed with standard Purina chow or fasted during 16 h following the standard guidelines and considerations for metabolic tolerance tests in mice [33], which have the approval of the institutional Animal Ethics Committee. To evidence the phosphorylation status of protein isoforms, the extracts were pretreated with bovine intestinal alkaline phosphatase as follows: 20 mg of protein in 100 mL of extract was incubated at 25 °C for 40 min with 1 mL (100 units) of enzyme. Under these conditions, this highly active enzyme of broad specificity digests all types of phosphate bonds [34,35,36]. Hence, proteins were resolved by resolutive PAGE/SDS followed by Western blot. 

### 2.5. Indirect Immunofluorescences

Cells were grown on collagenized coverslips, fixed with methanol at −20 °C for 10 min, and incubated overnight at 4 °C with 1/100 dilution of primary antibody and 1 h at room temperature with 1/200 dilution of the proper secondary antibody. Coverslips were mounted in a glycerol-based media with an anti-fade solution. Confocal microscopy images were acquired with a Nikon Eclipse-E800 confocal microscope using a Nikon DSU1 camera with ACT-2U software. 

### 2.6. Molecular Modeling and Structural Analysis 

FKBP51 three-dimensional protein structural models were built by homology modeling using the Swiss Model workspace [37] and Phyre2 web server [38] (http://www.sbg.bio.ic.ac.uk/phyre2) accessed on 5 January 2019. In both web servers, the protein sequence >gi|83404904|gb|AAI11051.1| hFKBP51 from NCBI database accessed on 5 January 2019 (http://www.ncbi.nlm.nih.gov/protein) was used as input. For the Swiss Model workspace, the human (1kt0.pdb) and the monkey structure of FKBP51 (1kt1.pdb) were used as templates. Both the human FKBP5 (5omp_A) and FKBP51 (1kt0_A) protein structures were modeled in the Phyre2 platform based on heuristics to maximize confidence, percentage identity, and alignment coverage, 44 residues were modeled by ab initio. The quality assessment of the models was performed by ERRAT [39] and Procheck [40] programs from SAVES 6.0 interactive validation server, Putative FK506 binding site in the models of hFKBP51 was determined by a structural alignment between hFKBP12-Fk506 complex (1fkf.pdb) and mFKBP51 (1kt1); and after making a residue correlation. Neighboring residues to those that shape the putative FK506 binding site were calculated by means Molmol program using a 5.0 Å cut-off, and their accessible surface area was calculated with the Surface Racer program with a probe radius of 1.5 Å.

### 2.7. Statistical Tests

All data shown in this study are the consequence of at least three repetitions of each experiment. Western blots are the most representative ones. Semi-quantifications of protein bands are shown as the mean value ± SD and were statistically analyzed according to the one-way nonparametric ANOVA examination followed by the Kruskal–Wallis test.

## 3. Results

### 3.1. Mitochondrial-Nuclear Shuttling of FKBP51

FKBP51 has recently been reported to localize to both the nucleus and mitochondria of cells, this subcellular localization being dynamically altered by cellular conditions [17,18,19,20,21]. Therefore, FKBP51 shuttles dynamically between the nuclei and mitochondria. Consistent with this concept, Figure 1a shows confocal microscopy images of endogenous FKBP51 expressed in L1-3T3 fibroblasts where the mitochondrial staining of FKBP51 (green) strongly colocalizes with the mitochondrial marker MitoTracker (red) and undergoes rapid full nuclear accumulation 20 min after cell exposure to hydrogen peroxide. These results confirm those previous reports. Interestingly, such relocalization of the immunophilin is reversible since the replacement of the culture medium by a peroxide-free medium rapidly restores the mitochondrial primary localization of FKBP51 within 20–30 min after peroxide washout. This indicates that the repopulation of the mitochondrial pool of the immunophilin due to de novo synthesis is unlikely. The suppression of the signal shown by cells transfected with a siRNA to knock down the expression of FKBP51 supports the specificity of the fluorescent signal seen in the other panels, as well as it being evidenced in previous works due to the use of different antibodies against the immunophilin such as the mouse MG19 clone, the rabbit antiserum R51, the rabbit polyclonal IgG from ABR, and the anti-*flag* M2 monoclonal IgG when cells being transfected with *flag*-tagged-FKBP51 (see later on). 

Figure 1b demonstrates that FKBP51 not only moves from mitochondria to the nuclei upon the onset of oxidative stress, but its expression is also rapidly induced by this stimulus. Likewise, the rapid and significant loss of FKBP51 level evidenced after peroxide wash out is compatible with an unusually high turnover of the protein, possibly due to a rapid proteasome degradation (see later). It should be remarked that this nuclear accumulation of the immunophilin does not appear to be exclusive for oxidative stress events since it has also been observed in other publications under other situations of relative stress such as cell confluence or when they are induced to differentiate (for example, L1 fibroblast into adipocytes [18,41]). Importantly, the relocalization of FKBP51 is also observed in vivo by mouse liver fractionation (Figure 1c). While FKBP51 distributes ubiquitously in normally fed mice, it accumulates in the nuclear fraction of fasted animals, presumably due to the metabolic stress generated by overnight fasting [33].

### 3.2. Phosphorylation of FKBP51

Resolutive 15% PAGE/SDS shows at least five FKBP51 isoforms in mouse liver extracts when the Western blot is developed with an anti-FKBP51 primary antibody. These bands were assigned to differentially phosphorylated variants of the immunophilin, as it is suggested by the effect of pretreatment of the extracts with bovine intestinal alkaline phosphatase (Figure 2a), a highly active phosphatase with a broad spectrum of substrates. To determine whether different isoforms may be located in a given subcellular compartment, cell lysates were fractionated into mitochondria, nucleus, and cytoplasm, and then resolved by Western blot. Figure 2b shows differences between each subcellular pool and between samples from fasted and fed animals. Note that isoform c prevails in mitochondria and is a minority in the cytoplasm, whereas the slower isoforms d and e are shown at higher amounts in the case of fasted animals and are absent in the cytoplasmic fractions. 

Figure 2c shows the relative abundance of phosphorylated forms when FKBP51 was immunopurified from different subcellular fractions of liver extracts. Interestingly, the most significant phosphorylated form of FKBP51 in all subcellular fractions is P-Thr, and its reactivity increases showing new bands of lower mobility in fasted tissues. In other words, these observations suggest that the stress generated by fasting may enhance phosphorylation in the threonine residues of the immunophilin.

### 3.3. Differential Phosphorylation of FKBP51 According to the Stimulus

In previous studies [15,31,42], we demonstrated that the treatment of undifferentiated cells of nervous lineage with the immunophilin ligand FK506 favors their differentiation into neurons. Microscopy studies showed that, like treatments with peroxide, the macrolide also favors the rapid translocation of FKBP51 from cytoplasm and mitochondria to the nucleus in undifferentiated neuroblastoma N2a cells (Figure 3a). Controlled treatments with alkaline phosphatase also indicate that FKBP51 may be resolved into five isoforms with different grades of phosphorylation (Figure 3b). Therefore, it was analyzed whether both stimuli, peroxide and FK506, may replicate the phosphorylation pattern of the immunophilin in addition to triggering its rapid accumulation in the nucleus. This was assayed using 3D-cultures of frontal brain slices exposed to 0.25 mM H_2_O_2_ or 1 μM FK506 (Figure 3d), a model that allows the evaluation of the response maintaining the organization of the nervous tissue with preservation of cytoarchitecture and cell populations. Inasmuch as brain slices from the frontal cortex are maintained in the plates during several days, the first step was the assessment of the cell death in the tissue by measuring the enzymatic activity of CPK, LDH, and GOT released into the medium. Figure 3c shows that after three days, the tissue tends to be normalized under the new culture conditions. Consequently, we decided to perform the assays the next day. Hence, brain slices were exposed for one hour to 0.5 mM H_2_O_2_ or 1 μM FK506, the tissue was homogenized, and proteins from the extract were resolved by Western blot (Figure 3d). Even though the basal pattern of expression of FKBP51 shows the same isoforms as those observed in L1-3T3 fibroblasts, the pattern of phospho-isoforms after the incubation is not alike when treatments with peroxide or FK506 are compared. Because the ideal separation of all five isoforms was not always achieved and band e is sometimes absent in an aleatory manner, we ran more replicable gels capable of resolving proteins into three stable bands that correspond to band a, band b, and bands c-d-e, and focused on these three main bands. Two FKBP51 bands normally fade after treatment and one is usually preserved, i.e., the fastest band *a* is preserved after peroxide and the middle band *b* is preserved after FK506. Both are reactive to an anti-phospho-tyrosine antibody. Therefore, in contrast to the enhancement of phospho-threonine residues observed in Figure 2 in mouse liver extracts, Tyr phosphorylation seems to be a predominant post-translational modification for activated FKBP51 in nervous cells.

### 3.4. FKBP51 Mutants

The experiments shown in Figure 2 and Figure 3 suggest that FKBP51 is influenced by phosphorylation. Note that conditions such as fasting, oxidative stress, or treatment with FK506 favor the predominance of an immunophilin that shows a different phosphorylation fingerprint. Based on the potential of amino acids such as Tyr, Thr, and Ser to be phosphorylated, the analysis of their accessible surface area to solvent, structural domain restrictions, the neighboring sequences, correlations between amino acid positions, etc., the potential phosphorylations of Tyr-54 and Thr-26, both located in the FK1 domain of the immunophilin (Figure 4a), were evaluated by mutations to alanine. Figure 4b is the expression control of the *flag*-tagged mutants compared to *flag*-wtFKBP51, which shows the same isoform profile as endogenous FKBP51 (see Figure 3d). The exogenous phospho-isoforms were evidenced by alkaline phosphatase treatment. Next, N2a cells transfected with these constructs were stimulated (or not) with peroxide as in Figure 3 to force the change of the isoforms’ pattern. Proteins were immunopurified with an anti-*flag* antibody and evidenced by Western blot (Figure 4c). Note that the *flag*-tagged purified isoforms mutated to alanine are not evidenced in the mutants’ constructs, thus confirming the prediction made by molecular modeling. 

Then, we asked whether the phosphorylation of those residues are relevant for the subcellular localization of FKBP51. The indirect immunofluorescence shown in Figure 5a demonstrates that the FKBP51 point mutants expressed in N2a cells localize in the same cell compartments as the wild-type FKBP51 (i.e., mitochondrial, cytosolic, and partially nuclear localization). When N2a cells were treated with peroxide, both mutants accumulated in the nucleus in similar fashion. The bar graph shows that there are no significant differences for the subcellular distribution of the mutants compared to wild type FKBP51, suggesting that at least these phosphorylations are not determinant for the redistribution of the immunophilin. The Western blot shows the basal expression of the three main endogenous isoforms of FKBP51 and the switch to the fastest migrated isoform when these cells are exposed to peroxide for different period of times. Note that FKBP51 is degraded upon long exposures times (4 h) with H_2_O_2_, this degradation likely to occur via proteasome as suggested by the protective action of the specific MG132 inhibitor. In short, oxidative stress translocates FKBP51 into the nuclei, the phosphorylation pattern of the immunophilin is affected, and the protein becomes unstable, such that it is targeted to proteasome degradation at longer incubation times.

## 4. Discussion

FKBP51 is an abundant and ubiquitously located Hsp90-binding immunophilin that plays vital roles in several systems. This chaperone is part of protein triages able to regulate the biological functions of several client proteins [43,44,45]. Like all proteins showing scaffold features, FKBP51 is a crucial regulator for many signaling pathways that function by interacting with multiple members of a pathway and tethering them into complexes. Consequently, changes in the subcellular distribution of FKBP51 and/or its turnover in the cell must impact all those functions and, at the same time, this represents an alternative therapeutic opportunity by targeting the proper client protein. In this study, it is demonstrated that the mitochondrial pool of FKBP51 rapidly migrates to the nucleus upon the onset of oxidative stress, a situation that mimics the metabolic state of cancer cells [21], and it is also observed when undifferentiated neuronal cells are stimulated with FK506, a ligand that triggers neurodifferentiation in vitro and in vivo conditions [42]. Interestingly, the experiment of Figure 1 indicates that such trafficking is reversible since the washout of peroxide restores the primarily mitochondrial localization of FKBP51, and given the short period of time involved (30 min), the possibility that the full appearance of FKBP51 in mitochondria is due to de novo synthesis and the simultaneous disappearance of the nuclear pool is due to protein degradation are both unlikely events. 

In this work, it is also demonstrated that tyrosine-54 and threonine-26 are functional targets of kinases that respond to specific stimuli. Both residues are located in a protruding and therefore solvent-exposed area of the FK1 domain. Interestingly, it seems that endocrine signaling is prone to modifying Thr-26 and FK506 more likely affects Tyr-54 phosphorylation. Whatever the favorite modification in each case, it is clear that both stimuli are not alike, suggesting specific post-translational modifications according to the type of stimulus. On the other hand, neither one nor the other phosphorylation affects the subcellular localization of FKBP51 or its shuttling capacity between organelles, whereas oxidative stress targets the remaining Tyr-54 isoform of FKBP51 to proteasomal degradation. 

It has recently been demonstrated that the mitochondrial Ser/Thr kinase Pink1 (PTEN-induced putative kinase 1) forms complexes with mitochondrial FKBP51 that are more abundant when cells are exposed to stress, this interaction being responsible for the phosphorylation of FKBP51 in still unidentified serine residues [22]. Like it is the case in our study for the specific phosphorylations in Thr-26 and Tyr-54, Pink1-dependent phosphorylation does not affect the mitochondrial localization of the immunophilin. Even though the NetPhos-3.1 prediction tool used to analyze the possible phosphorylation sites of FKBP51 provided a high potential score for Thr-26 and Tyr-54, it was unable to assign a possible kinase for these sites (both are tagged as “unspecific kinase”). Nonetheless, during the progress of the present study, it was published that FKBP51 could be a bona fide substrate of SRMS (Src-related kinase lacking C-terminal regulatory tyrosine and N-terminal myristylation sites), a tyrosine-kinase that shows as a target that Tyr-54 of FKBP51 [46]. Such phosphorylation disrupts the interaction of FKBP51 with PHLPP (PH domain leucine-rich repeat protein phosphatase) and consequently, prevents PHLPP-dependent dephosphorylation of AKT, which causes sustained AKT activation and uncontrolled cell growth. In this regard, the nuclear localization of FKBP51 observed in most cancer cells (rather than the cytoplasmic/mitochondrial localization seen in normal cells) [21] is perhaps the direct consequence of the high production of ROS in these cells, a stimulus that is tightly linked to the enhanced catalytic activity of protein kinases [47,48,49,50].

In cancer cells, there is also a great enhancement of the telomerase activity due to the recruitment of nuclear FKBP51 to hTERT catalytic subunit [21]. In line with this advantageous effect of the immunophilin for cancer cell survival and cancer cell division, it has also been reported the presence of FKBP51 in nuclear heterocomplexes of Amotl2 (angiomotin-like 2) and IQGAP1 (IQ-motif-containing GTPase-activating protein 1), factors that are involved in the regulation of cell growth, tumorigenesis, inflammation, immunity, cell plasticity and differentiation [30]. Furthermore, nuclear FKBP51 is also recruited to the promoter region of NF-kB target genes [14], and it has been suggested that the immunophilin may also participate in the transcriptional events that govern cyclin D1 oscillations during the cell cycle [51]. It has been proposed that the nuclear rather than a mitochondrial or cytoplasmic localization for FKBP51 strongly depends on the involvement of its TPR domain since TPR mutants unable to interact with Hsp90 show a constitutively nuclear FKBP51 [17], and it is the case for a spliced variant lacking the TPR domain that has been found in melanoma patients [52]. 

Treatments with macrolide FK506 result in a specific pattern of certain phosphorylated isoforms of FKBP51 where phosphorylation in serine and threonine residues are lost, whereas a phosphorylation of tyrosine is preserved. Inasmuch as this pattern is different from that evidenced after exposure to peroxide, a specific functional role for each phosphorylated isoform of FKBP51 according to the type of stimulus can be inferred. Notably, fibroblast differentiation into adipocyte does not show an equivalent pattern to neuronal differentiation, but a functional regulation by PKA [18]. An analysis for predicted phosphorylation sites by PKA indicated a high possibility for a conserved Ser-312, whereas molecular modeling analysis also predicted the presence of two putative PKA phosphorylation sites in Ser-160 and Thr-285 [18]. These potential sites are currently under investigation. 

One of the benefits of a cell using scaffold proteins such as FKBP51 is the ability of these hub proteins to organize signaling complexes by specific protein-protein interactions that can affect the pathway function. They are easily regulated by external signals that can modify the association of other proteins with the scaffold. Post-translational modifications add great variability to those modifications; thus, phosphorylations at specific residues are responsible for protein activation, relocalization to particular compartments, protein degradation, and acquisition of other biological functions. This can be used as a target to pharmacologically regulate those processes. Moreover, like most members of the immunophilin family, FKBP51 adds the possibility of generating conformational changes in the client factor by proline isomerization. Therefore, both its scaffold capability and the enzymatic intrinsic activity of peptidylprolyl isomerase are complementary properties. Nonetheless, it is still uncertain whether and up to what level one property, i.e., the phosphorylation of a specific amino acid residue located in the FK1 domain (such as the post-translational modifications evidenced in this study) could affect the potential isomerase activity of FKBP51 on a given client protein. 

## Figures and Tables

**Figure 1 cells-11-03771-f001:**
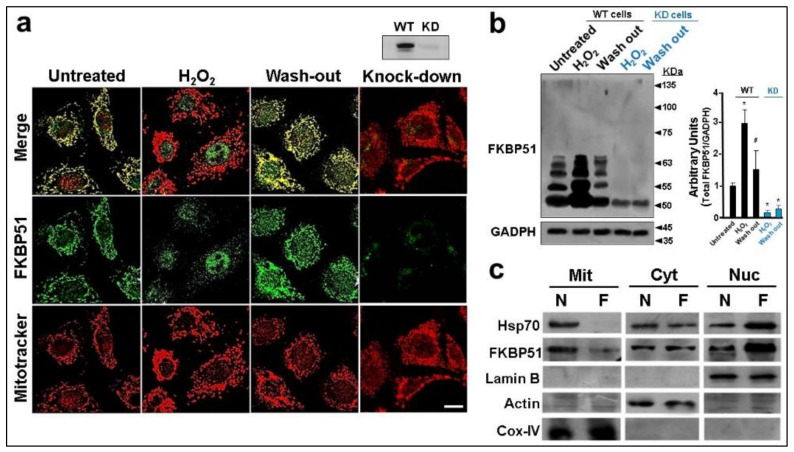
FKBP51 undergoes nuclear-mitochondrial shuttling. (**a**) Confocal microscopy of 3T3-L1 fibroblasts stained for the mitochondrial marker MitoTracker (red) and wild type (WT) FKBP51 (green). FKBP51 translocates to the nucleus after 20 min of cell exposure to 0.25 mM H_2_O_2_, and fully cycles back to mitochondria after 30 min the peroxide was washed out. The knock-down (KD) of FKBP51 fully abolishes the signal (Western blot shown on the top of the column). White bar = 10 um. (**b**) Western blots for endogenous FKBP51 were performed in lysates from untreated cells, after 1 h of cell exposure to H_2_O_2_, or 1 h after the peroxide was washed out. Blue labels correspond to cells treated with siRNA against FKBP51 (KD). The plot shows a semi-quantification of the bands by densitometry (* different from untreated cells at *p* < 0.005; # wash out WT cells versus peroxide-treated WT cells, different at *p* < 0.010). (**c**) Biochemical fractionation into mitochondria (Mit), cytosol (Cyt), and nuclei (Nuc) of liver extracts from normally fed mice (N) and mice fasted overnight (F). Cox-IV, actin, and lamin B were used as markers of each respective subcellular fraction.

**Figure 2 cells-11-03771-f002:**
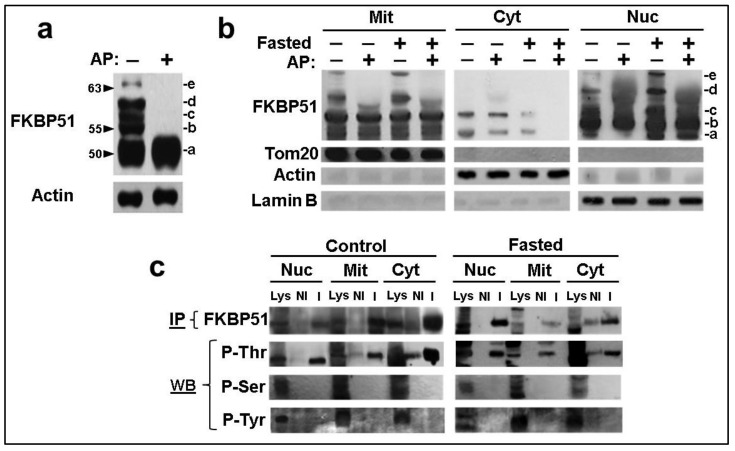
FKBP51 is a phosphoprotein. (**a**) FKBP51 isoforms were resolved in a 15%PAGE/SDS followed by Western blot after incubation of mouse liver extracts for 40 min at 30 °C without (−) and with (+) alkaline phosphatase (AP). The FKBP51 isoforms are labeled with letters (a–e). (**b**) FKBP51 isoforms (a–e) were resolved in mitochondria (Mit), cytosol (Cyt), and nuclei (Nuc) after biochemical fractionation. Tom20 was used as a mitochondrial marker, actin as a cytoplasmic marker, and lamin B as a nuclear marker. (**c**) After biochemical fractionation of control and fasted mouse livers, FKBP51 was immunoprecipitated (IP) and Western blotted (WB) with the indicated anti-phospho-amino acid antibodies. Lys = Whole lysate. NI = Non-immune IgG. I = IgG against FKBP51.

**Figure 3 cells-11-03771-f003:**
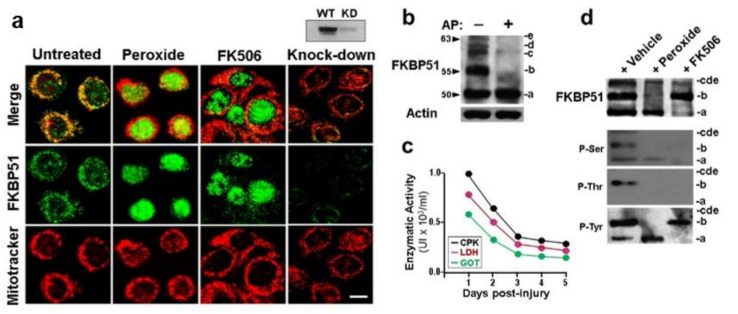
The phosphorylated isoforms of FKBP51 are modulated according to the type of stimulus. (**a**) Nuclear relocalization of FKBP51 (green) in N2a cells treated with H_2_O_2_ as in Figure 1 or with 1 μM FK506 for 4 h. Mitochondria are stained with Mitotracker (red). The insert shows a Western blot demonstrating the efficiency of the knockdown (KD) compared to the wild-type extract (WT). (**b**) Alkaline phosphatase (AP) treatment evidence up to five FKBP51 phospho-isoforms in N2a cells (named with letters). The MW markers are shown next to the blot. (**c**) Residual enzymatic activity in the medium. (**d**) The stripping membrane of the Western blot for FKBP51 shown on the top demonstrates that the relative abundance of each phospho-amino acid. (a–e) is differentially affected when 3D frontal cortex slices are treated with H_2_O_2_ or FK506.

**Figure 4 cells-11-03771-f004:**
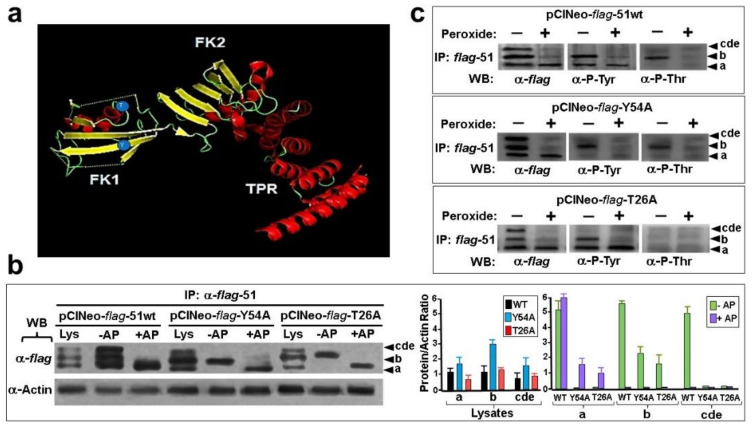
Tyr-54 and Thr-26 are phosphorylation targets of FKBP51. (**a**) Showing Tyr-54 and Thr-26 as blue spheres. Both residues were point-mutated to Ala. (**b**) N2a cells were transfected with the *flag*-tagged constructs, FKBP51 was immunoprecipitated (IP) with anti-*flag* antibody, and the respective pellets were digested with alkaline phosphatase (AP). The bar graph shows the densitometry for each band (*n* = 3, mean ± SD). Lys = Whole lysate. (**c**) Transfected N2a cells were treated with H_2_O_2_ as in Figure 3, FKBP51 was immunopurified with anti-*flag* antibody, and pellets resolved by 15% PAGE/SDS followed by Western blot (WB) with α-*flag* or α-P-Tyr and α-P-Thr), as indicated. The nomenclature of the isoforms is the same as in the other figures. (a–e) is differentially affected when 3D frontal cortex slices are treated with H_2_O_2_ or FK506.

**Figure 5 cells-11-03771-f005:**
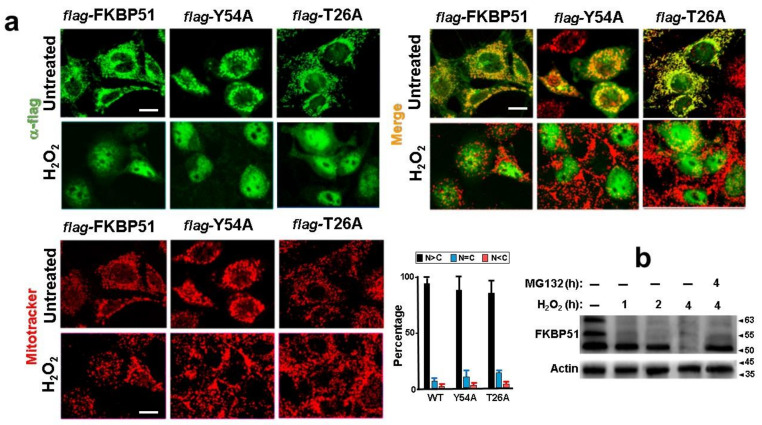
(**a**) The subcellular localization of FKBP51 is not affected by Tyr-54 and Thr-26 phosphorylation. Confocal microscopy of transfected N2a cells with *flag*-tagged FKBP51 mutants and *flag*-tagged FKBP51 wild type. Basal and the peroxide-dependent nuclear localization of both FKBP51 mutants and wild-type immunophilin are alike (green). Mitochondria were stained with Mitotracker (red). Bar = 10 um. The bar graph shows no significant differences for the subcellular distribution of the mutants. (**b**) Peroxide treatment targets FKBP51 to proteasome degradation. N2a cells were treated with 0.25 mM H_2_O_2_ for 1 h, 2 h, or 4 h, and FKBP51 was Western blotted with the commercial rabbit polyclonal IgG. Endogenous FKBP51 is dephosphorylated and degraded after 4 h, presumable due to proteasome degradation since degradation is prevented with 10 μM MG132.

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
