# Peer review of "The Scaffold Immunophilin FKBP51 Is a Phosphoprotein That Undergoes Dynamic Mitochondrial-Nuclear Shuttling"

_cells, 2022, doi:10.3390/cells11233771_

Round 1
Reviewer 1 Report
Zgajnar et al. describe the characterization of the immuophilin FKBP51, a cochaperone of Hsp90 that recently has received considerable attention especially in stress biology and as a risk factor for stress-related disorders. The current manuscript deals with two molecular aspects of FKBP51: (i) subcellular localization, especially mitochondrial-nuclear shuttling, and (ii) potential regulation of FKBP51 by phosphorylation.
The first aspect, mitochondrial-nuclear shuttling, has been investigated and described by the author in several previous publications and the novelty-factor here appears to be rather limited. The second aspect, putative FKBP51 phosphorylation site, are novel, which could potentially be very important. Unfortunately, the work suffers at several levels from substantial technical flaws and shortcomings:
1. Show all results alluded to in the text as ‘data not shown’
2. The western blots are absolutely critical for this manuscript. Please show as supplementary information the full Western blots including size markers. An indication of size is absolutely critical to get an estimation of the size shift induced by the putative phosphorylation
3. Fig. 1a: Show a Western blot, which confirms the siRNA knockdown of FKBP51
4. Repeat the experiment in Fig. 1b with and without siRNA knockdown to confirm the specificity of the antibody for the higher induced bands.
5. Fig. 3a: Include a knockdown by siRNA for the localization experiments in N2a cells as in Fig. 1a for 3T3-L1 cells (incl. the confirmation of the knockdown by WB)
6. Fig. 3d: Why did the authors use FK506? There are now much more selective FKBP51 inhibitors commercially available, e.g. https://www.caymanchem.com/product/24366
7. The experiment in Fig. 4 is absolutely critical but highly inconclusive at present. E.g. how do the authors explain the existence of the identical set of additional bands in wtFKBP51 as well as both mutants? How do they explain a putative faster moving band for the putantive pY54-posphorylated protein.
Please repeat this experiment and include:
a) a side-by-side mock transfection followed by identical processing (IP & WB) to show the specificity of the procedure (for experiments 4b-d)
b) Include lanes with or without alkaline phosphatase treatment (for experiments 4b-d)
c) Please show FLAG-western blots form the input lysates (for experiments 4b-d), using the same gel system. Importantly, include a side-by-side mock transfection as specificity control.
d) Please repeat this experiment (with FLAG-FKBP51 overexpression followed by FLAG-WB analysis) in a second cell line, e.g. 3T3-L1 cells
8. In Fig. 5a, please show separate images for FKBP51-stains and MitoTracker, in addition to the merge pictures. Importantly, include a side-by-side mock transfection as specificity control. How was FKBP51 stained. I assume by a Flag-antibody. If not repeat this experiment with a anti-FLAG-antibody.
9. In Fig. 5b, these were untransfected N2a cells, correct? How was the staining for FKBP51 performed?

Author Response
Ref: Cells-1932485: The Scaffold Immunophilin FKBP51 is a Phosphoprotein that Undergoes Dynamic Mitochondrial-Nuclear Shuttling.
Authors: Nadia R. Zgajnar , Cristina Daneri-Becerra , Ana Cauerhff , Mario D. Galigniana
Dear Dr. Theodora Banu,
Thank you for your comments and those by the reviewers. All concerns that were possible to address were answered and properly remarked on in the amended version. Version 2 shows new experiments and new and modified figures. We would like to acknowledge and thank the reviewers for their comments since several observations have helped us to improve the overall presentation of this article. The specific answers are as follows:
REVIEWER 1:
R= The first aspect, mitochondrial-nuclear shuttling, has been investigated and described by the author in several previous publications and the novelty-factor here appears to be rather limited.
A= The fact that FKBP51 a) is mitochondrial, and b) undergoes mitochondrial-nuclear shuttling is not a widely known feature of the immunophilin, such that most people are unaware of this property. The best evidence is that controls already done and published such as antibody specificity or siRNA assays are requested. Nonetheless, these controls have been included in this new version.
- Show all results alluded to in the text as ‘data not shown’.
A= Correct, today’s style is to show the figures as supplementary material. There were four “data not shown” statements in the original text, and they referred to: (1) siRNA and band specificity= the WBs were included in the new version; (2) phosphorylated isoforms of FKBP51 in N2a cells were equal to those shown in brain extracts, and the profiles of N2a cells were already shown in Figs.4 and 5b of the original version and are also shown in the new Fig.4; therefore the observed statement was deleted; (3) Ditto for SH-SY5Y cells. Since the WBs are exactly the same as those shown for N2a and brain slices, this statement was also deleted from the text; (4) “other types of stress also trigger nuclear accumulation of FKBP51” = we do not wish to show these figures in this paper because there is a manuscript in preparation related to the role of HSF1 in the process, but attached some pictures as a piece of evidence to be seen exclusively by the reviewers (Mitotracker: red; FKBP51: green)=
- The western blots are absolutely critical for this manuscript. Please show as supplementary information the full Western blots including size markers. An indication of size is absolutely critical to get an estimation of the size shift induced by the putative phosphorylation.
A= Correct. New Figure 1-b shows a whole gel and the proper MW markers were included (see the original scan of the film attached below). Note that we named each isoform with a letter to simplify the nomenclature of the figure, and then we label the bands accordingly rather than with the MW. Also, note that highly resolutive gels (15%, overnight at 10 mA) may resolve up to 5 isoforms, whereas regular runnings (12% at 80 mA) do not resolve the upper bands (named cde). In most of the gels we pragmatically concentrated on the three most relevant ones.
Original:
- Fig. 1a: Show a Western blot, which confirms the siRNA knockdown of FKBP51
A= In the revised version, IIFs shown in Figs. 1-a (L1-3T3 cells) and 3-a (N2a cells) are showing a WB on the top of the proper column.
- Repeat the experiment in Fig. 1b with and without siRNA knockdown to confirm the specificity of the antibody for the higher induced bands.
A= The new gel shown in Fig.1-b shows the requested condition.
- Fig. 3a: Include a knockdown by siRNA for the localization experiments in N2a cells as in Fig. 1a for 3T3-L1 cells (incl. the confirmation of the knockdown by WB).
A= Done.
- Fig. 3d: Why did the authors use FK506? There are now much more selective FKBP51 inhibitors commercially available, e.g. https://www.caymanchem.com/product/24366.
A= Our studies related to the role of Hsp90-binding immunophilins in the processes of neurodifferentiation and neuroregeneration were conducted with FK506. The findings demonstrated that the final effect is the result of the balance of FKBP51 versus FKBP52 activation by the drug, like a ying-yang action since FK506 shows equivalent Ki (ranging from 100 to 200 nM) for both immunophilins and both are affected by the doses used in the assays. Our interest is focused on the putative mechanisms of action of the FKBP-drug complex. One of the preliminary results suggested a differential activation of FKBP51. Perhaps SaFiT (specific for FKBP51) could show similar actions, but first, we wish to determine the modifications exerted by the drug used during all these experiments. SaFiT could be assayed for comparative purposes in the future when we get some answers with FK506 to the present question.
- Theexperiment in Fig. 4 is absolutely critical but highly inconclusive at present. E.g. how do the authors explain the existence of the identical set of additional bands in wtFKBP51 as well as both mutants? How do they explain a putative faster moving band for the putantive pY54-posphorylated protein. Please repeat this experiment and include: a) a side-by-side mock transfection followed by identical processing (IP & WB) to show the specificity of the procedure (for experiments 4b-d) b) Include lanes with or without alkaline phosphatase treatment (for experiments 4b-d) c) Please show FLAG-western blots form the input lysates (for experiments 4b-d), using the same gel system. Importantly, include a side-by-side mock transfection as specificity control. d) Please repeat this experiment (with FLAG-FKBP51 overexpression followed by FLAG-WB analysis) in a second cell line, e.g. 3T3-L1 cells
A= The antibody does not recognize specific phospho-FKBP51 isoforms, and there are several potential residues feasible to undergo not only phosphorylations, but other post-translational modifications. We do not know what they are, the target residues, how they migrate, and whether a specific running condition is able to resolve them or not. Therefore, the only truth is the reality of the experimental facts, these are the bands observed under these conditions. The important information here is the comparative differences evidenced between wt-51 and the mutants in whole lysates when antibodies against specific phospho-amino acids are assayed. Moreover, note that the new IP with anti-flag IgG clears all doubts related to the pattern observed with whole lysates, such that only one band is now revealed for each transfected mutant. In the revised version, the exogenous FKBP51s were IP with anti-flag IgG and the specificity of the WB for the isoforms was also evidenced by alkaline phosphatase digestion, as the reviewer wisely suggested. The original of this film is attached below. Because this experiment is the one that shows the lowest number of repetitions (see Reviewer 1 comments), next to the WB is shown a semiquantitative bar graph plot.
- In Fig. 5a, please show separate images for FKBP51-stains and MitoTracker, in addition to the merge pictures. A= Done.
Importantly, include a side-by-side mock transfection as specificity control. A= If the reviewer observes both the original IIF and the new one (the untreated condition shows now better cells with clearer Mitotracker staining), one evident fact emerges= in the same field there are cells that do not express FKBP51 (only red staining). These are the best possible controls, “mock-like” cells in the same field as those efficiently transfected.
How was FKBP51 stained. I assume by a Flag-antibody. If not repeat this experiment with a anti-FLAG-antibody. A= Correct, this is why “only red” cells are seen in the field.
- In Fig. 5b, these were untransfected N2a cells, correct? How was the staining for FKBP51 performed?
A= This is endogenous FKBP51, it was revealed with the ABR antibody. This was clarified in the legend of the revised version.
REVIEWER 2:
This overall is a nicely written paper, which addresses phosphorylation and compartmental translocation of FKBP51 in various test systems. The experiments shown seem well performed, include the necessary controls and the results are interesting from a protein biochemistry point-of-view. However, the following major points seriously dampen my enthusiasm: (1) Only representative results are shown and the number of independent experiments performed is not given. (2) Several results are mentioned in the results section without giving the supportive data ("data not shown") which in my view is not acceptable anymore. Overall, these two points make it impossible to assess the robustness of the data.
A= See answers to Reviewer 1. Regardless of the number of repetitions (now mentioned in Methods), it should be pointed out that just in this manuscript the resolution of FKBP51 isoforms is evidenced seven times (Figs. 1-b, 2-a, 2-b, 3-b, 3-d, 4-b, and 4-c), without counting the number of repetitions of each panel simply to show the best WB.
Finally, the authors use mouse samples including mice on a fasting diet but do not mention anything about these mice in the Material and Methods Section, not even information on ethical approval for these animal studies.
A= This information was included in section 2.4 of Methods. In this version, we have included a mention of the ethical approval for the overnight fasting procedure.
MAJOR COMMENTS: 1) Unfortunately, the authors do not provide any information on number of independent experiments performed and the data shown is restricted to single representative experiments without quantification. Therefore, data reproducibility and robustness of the results shown cannot be assessed.
A = Already answered.
2) Ideally, the authors should have performed at least three independent experiments and, at least for key experiments, provide an objective measure of the data by quantification of bands/IF signal and proper statistical analysis. Otherwise, none of the data shown is convincing enough.
A = Note that a Western blot is not a quantitative method, but a qualitative one. A protein band is present or absent, and its expression is relatively increased or decreased (compared horizontally in the same gel), but not a quantitative measurement since the signal is related to, but not directly proportional to the quantity of protein. Additionally, the detection system saturates early the signal underestimating its amount. However, band scanning is often useful to SEMI-QUANTIFY the bands and demonstrate that a given pattern of proteins is the same every time proteins are resolved and not the product of a single gel, which is not our case. As stated before, just in this manuscript the resolution of FKBP51 isoforms is evidenced seven times, independently of the number of required repetitions due to technical reasons. Nonetheless, there is one new experiment that was required by Reviewer 1 where the semi-quantification of the bands is required since there were no previous essays like this one and the experiment was performed only three times: Fig. 4-b. Here it is important to compare +AF vs -AF to demonstrate the specificity of the phosphorylated bands by digestion with the phosphatase.
MINOR COMMENTS:
- Line 210: The authors state that nuclear FKBP51 accumulation “is not exclusive for oxidative stress events, but also observed (not shown here) under any situation of relative stress (i.e.: nutrient deprivation, oxidants, chemicals, heat, cold, UV light, chemicals, etc.)”. This statement should be either supported by additional data, e.g. in the supplement, or removed if the authors decide to not show the data.
A= Already answered (see above, reviewer 1). Note that the attached figure is supporting material for revision purposes only since it is part of a manuscript in preparation.
- Line 267: Similarly, the authors state that “similar results were obtained with SH-SY5Y neuroblasts (not shown).” Again, I recommend the authors to refrain from stating results without showing the supporting data.
A= Agree. The sentence was deleted.
- Line 272: Along the same lines, the statement “This was first assayed in N2a cells (not shown) as described in Figure 1 for L1-3T3 fibroblasts” should be removed or the supporting data shown. (Here, however, I am confused because the authors actually show such N2a cell experiments in Figure 3a and b? Is “data not shown” actually true or, if not, what is the difference between the data shown and the data not shown?).
A= Agree. The sentence was deleted.
- In addition to major comment 1: Line 239: The authors state that “Figure 2B shows quali-quantitative differences”, but the changes shown remain merely qualitative without quantification.
A= Agree. The sentence was deleted. We attempted to “normalize” the scans of these bands in several manners, but the number of isoforms in each organelle where the standard protein varies made it difficult to find a rational comparative method for each subcellular fraction. Therefore, we decided to delete the sentence.
- Can the authors please comment on whether bovine intestine alkaline phosphatase (type VII-NT) dephosphorylates all different protein phosphorylations (i.e. P-Thr, P-Tyr, P-Ser)?
A= This phosphatase isolated from the bovine intestine is highly promiscuous since it shows broad specificity of substrate and hydrolyse all types of phosphorylated compounds and proteins. As such, it is widely used for this purpose. See Fernley, H.N., "Mammalian Alkaline Phosphatases", in The Enzymes (Boyer, P.D., ed.), Vol. IV ("Hydrolysis"), pp. 417-447 (1971). In the revised version of this manuscript, we mention its high activity and broad specificity.
- Figure 5a: The representative IF figure suggests that both Y54A and T26A variants lead to comparably more nuclear FKBP51 while the authors state that “Basal (…) nuclear localization of both FKBP51 mutants and wild type immunophilin are alike”. This conclusion is not supported by this IF image. Again, some objective assessment of data derived from several independent experiments would help to strengthen (or disprove) the authors’ conclusion.
A= In the revised version a quantification plot was included (Fig.5) confirming that there are no differences between all conditions. This is somehow disappointing for us because we bet that this was going to be the case. Actually, this was one of the original aims of this study. Unfortunately for our expectations, data are data and showed that this is not the case.
- Small textual changes and corrections to grammar are needed, in particular the figure legends contain many typos. See some examples below (probably not a comprehensive list):
- Abstract: “fasting and oxidants stimulate FKBP51 translocation”
- Figure legend of figure 1: correct “mitocondrial”
- Figure legend of figure 1: I am not a native English speaker myself but to the best of my knowledge, there is no hyphen between particle and verb when you use phrasal verbs like “wash out” or “cycle back”.
- Figure legend of figure 3: correct “differentialy” and “FKBP51 phospho-isofroms”
- Figure 3 is missing the information on the scale bar.
- Figure legend of figure 5: correct “subcelular”, “targtets” and “Undergoes”
A= Thank you very much for the kind help of the reviewer. The texts were revised and modified accordingly. We do apologize for this inconvenience.
- Discussion: Line 372: -> plays; Line 399: It has recently been demonstrated; Line 403/404: phosphorylation
A= Amended. Thank you!

Reviewer 2 Report
Summary
This overall is a nicely written paper, which addresses phosphorylation and compartmental translocation of FKBP51 in various test systems. The experiments shown seem well performed, include the necessary controls and the results are interesting from a protein biochemistry point-of-view. However, the following major points seriously dampen my enthusiasm: (1) Only representative results are shown and the number of independent experiments performed is not given. (2) Several results are mentioned in the results section without giving the supportive data ("data not shown") which in my view is not acceptable anymore. Overall, these two points make it impossible to assess the robustness of the data.
Finally, the authors use mouse samples including mice on a fasting diet but do not mention anything about these mice in the Material and Methods Section, not even information on ethical approval for these animal studies.
Major Comment
Unfortunately, the authors do not provide any information on number of independent experiments performed and the data shown is restricted to single representative experiments without quantification. Therefore, data reproducibility and robustness of the results shown cannot be assessed. Ideally, the authors should have performed at least three independent experiments and, at least for key experiments, provide an objective measure of the data by quantification of bands/IF signal and proper statistical analysis. Otherwise, none of the data shown is convincing enough.
Minor Comments
1. Line 210: The authors state that nuclear FKBP51 accumulation “is not exclusive for oxidative stress events, but also observed (not shown here) under any situation of relative stress (i.e.: nutrient deprivation, oxidants, chemicals, heat, cold, UV light, chemicals, etc.)”. This statement should be either supported by additional data, e.g. in the supplement, or removed if the authors decide to not show the data.
2. Line 267: Similarly, the authors state that “similar results were obtained with SH-SY5Y neuroblasts (not shown).” Again, I recommend the authors to refrain from stating results without showing the supporting data.
3. Line 272: Along the same lines, the statement “This was first assayed in N2a cells (not shown) as described in Figure 1 for L1-3T3 fibroblasts” should be removed or the supporting data shown. (Here, however, I am confused because the authors actually show such N2a cell experiments in Figure 3a and b? Is “data not shown” actually true or, if not, what is the difference between the data shown and the data not shown?)
3. In addition to major comment 1: Line 239: The authors state that “Figure 2B shows quali-quantitative differences”, but the changes shown remain merely qualitative without quantification.
4. Can the authors please comment on whether bovine intestine alkaline phosphatase (type VII-NT) dephosphorylates all different protein phosphorylations (i.e. P-Thr, P-Tyr, P-Ser)?
5. Figure 5a: The representative IF figure suggests that both Y54A and T26A variants lead to comparably more nuclear FKBP51 while the authors state that “Basal (…) nuclear localization of both FKBP51 mutants and wild type immunophilin are alike”. This conclusion is not supported by this IF image. Again, some objective assessment of data derived from several independent experiments would help to strengthen (or disprove) the authors’ conclusion.
6. Small textual changes and corrections to grammar are needed, in particular the figure legends contain many typos. See some examples below (probably not a comprehensive list)
1. Abstract: “fasting and oxidants stimulate FKBP51 translocation”
2. Figure legend of figure 1: correct “mitocondrial”
3. Figure legend of figure 1: I am not a native English speaker myself but to the best of my knowledge, there is no hyphen between particle and verb when you use phrasal verbs like “wash out” or “cycle back”.
4. Figure legend of figure 3: correct “differentialy” and “FKBP51 phospho-isofroms”
5. Figure 3 is missing the information on the scale bar.
6. Figure legend of figure 5: correct “subcelular”, “targtets” and “Undergoes”
7. Discussion: Line 372: -> plays; Line 399: It has recently been demonstrated; Line 403/404: phosphorylation
Author Response

(The authors gave the same response as above.)

Round 2
Reviewer 1 Report
The authors have largely adressed the pending issues and the presented data are now much more credible.
Sometimes, the labelling of the Figures is a bit confusing (e.g., new Fig. 4b: bar on top indicates IP for all samples, but then Lysates are partially shown below, as well as actin bands. I assume these are form the samples before performing the IP).
Overall, however, the data now apear solid.
Reviewer 2 Report
All my comments have been satisfactorily addressed.